# Open Lesson as a Means of Teachers' Learning

Mustafa Abdulbakioglu [1], Anar Kolushpayeva [2], Nuri Balta [1], Nursultan Japashov [1,3,4,*] and Christine L. Bae [5]

1 Department of Pedagogy of Natural Sciences, Suleyman Demirel University, Almaty 040900, Kazakhstan
2 School of Politics and Law, Almaty Management University, Almaty 050060, Kazakhstan
3 Solid State Physics and Nonlinear Physics Department, Al-Farabi Kazakh National University, Almaty 050040, Kazakhstan
4 Physics Department, Nazarbayev Intellectual School of Chemistry and Biology, Almaty 050006, Kazakhstan
5 School of Education, Virginia Commonwealth University, Richmond, VA 23284, USA
* Correspondence: nursultandzhapashov@gmail.com

**Abstract:** Open lesson (OL), similar to the Japanese lesson study in many aspects, is a professional development model regularly used in schools in the Commonwealth of Independent States Countries. The purpose of this study was to examine teachers' and students' attitudes and beliefs about OL practices and activities, using both quantitative and qualitative methods. The sample consisted of 72 instructors and 239 university students in Kazakhstan. MANOVA results from teacher data and student data indicated weak positive attitudes and beliefs about OL. Moreover, no differences were found between the gender, discipline, nationality, and teaching experience of teachers, while students learning social sciences showed stronger positive attitudes and beliefs about the effect of OL on learning when compared to students learning natural sciences. Slightly positive attitudes and beliefs of participants imply that OL implementations in schools should be reconsidered.

**Keywords:** open lesson; teaching practice; teacher professional development; lesson study





## 1. Introduction

Open lesson (OL) is a kind of lecture conducted by qualified teachers to show and disseminate the most effective methods of instruction. Such lectures contribute to the implementation of the achievements of pedagogical science in practice, the dissemination of pedagogical experience, and the further training of teachers. OL is a model used by the educational program of the Republic of Kazakhstan and the Commonwealth of Independent States Countries (CISCs) and can serve as an important approach to education reform for CISCs. It is similar to the public research lesson method used in the USA [1], Japanese open lesson [2], or Chinese open class [3,4], and can be used as the vehicle to examine pedagogical practices and peer-to-peer teaching. In European countries, the closest equivalent to OL is the demonstration class [5]. Common to all of these professional learning models is the focus on students' thinking as teachers collaborate on a common learning goal, observe one another's teaching, and make targeted adjustments to their curriculum and pedagogy based on the evidence collected from their classrooms [6].

OL applications are mandatory for teachers from the primary level to the university level, especially for CISCs. Teachers typically see OL activities as an opportunity to exhibit their educational skills. Administrators also motivate teachers to conduct OL [7]. Although there is a wide usage of OL in many CISCs, related research is pretty rare. Existing studies in CISCs focus mainly on the application of the educational approaches in OL [8–10]. Similarly, worldwide studies focus on some aspects of OL, such as the stages and procedures of OL [3,4] and the effectiveness of OL [2,11,12]. Since related literature is rather rare, we conducted a study to investigate teachers' and students' attitudes and beliefs about OL. We studied attitudes and beliefs to expand on the literature and to attract policymakers' attention to the implementation of OL in schools. Attitudes

and beliefs are important constructs to examine, as there is a large body of literature showing that these constructs predict sustained commitment to the implementation of new pedagogies for teachers and persistence in learning for students [13–15]. Demographic variables, such as gender, age, ethnicity, grade, experience, discipline, and numerous others, are common in many educational types of research. For example, Yan [16] studied the relationship between teachers' beliefs regarding self-regulated learning (SRL), together with key demographic variables, including gender, school sector, and teaching experience, and their SRL instruction.

The theme of this study may be interesting to researchers on two points. First, it is interesting to see the existence and the detail of such teachers' practices (open lesson) in different countries. Up to now, only several specific countries have reported some practices of OL (Japan, China, the USA, European countries, etc.), while those of other countries have not. Second, a study like this allows researchers to further ask interesting questions on the differences in similar practices between countries, on the conditions or affordance for implementing such teachers' practices in different countries or for scaling up or improving them, and on the constraints that hinder such practices.

The following research questions guided this study:

- What are the beliefs and attitudes of teachers and students towards OL?
- Do the beliefs and attitudes differ across gender, teaching experience, age, nationality, and discipline groups for teachers and students?
- What are the advantages and disadvantages of OL for teaching and students' learning?

## 2. Literature Review

### 2.1. Aim of Open Lesson

The aims of using OL include demonstrating improved or advanced forms and methods of instruction, analyzing the effectiveness of teaching aids to generalize methods of scientific organization, and monitoring the quality of the learning process [3].

The goals of the teacher, when conducting an OL class, can be to evaluate the effectiveness of the applied technologies and methods, improve individual techniques and pedagogical findings, and/or test his/her new approaches to teaching and educational work. The goal of the lesson is pre-determined by the teacher. The goal of an OL class could be divided into several parts: (a) educational goals (formation of practical experience and a systematic scientific knowledge among students), (b) development goals (development of cognitive processes of students, professional skills, and personal qualities), and (c) training goals (the formation of personality and character among students) [11,17].

Moreover, in the case of pre-service teachers, OL can also help them develop a professional understanding of their role by deepening their understanding of students' learning difficulties and helping them to understand the real situation in the classroom [12].

### 2.2. Application of Open Lesson

The structure of OL is quite simple: A group of teachers observes a lesson taught by a colleague and discuss its pros and cons right after the lesson by analyzing various aspects, such as the goals of the lesson, merits and weaknesses, activities completed during the lesson, and shortcomings and mistakes made in the organization and content of the OL. Besides schoolteachers, there may also be other invited experts, such as college and university teachers, who participate in the lesson and discussion [18]. For an OL class, the teacher prepares a full set of documents that detail the pedagogical plans and methodologies used in the lesson, which generally includes a study plan, lesson plan, concept of the lecture, set of materials and handouts, tasks for independent work, set of multimedia materials, and/or tasks or questions for independent extracurricular work [19].

Usually, an OL class lasts for one period (40–45 min) of teaching. The topic strictly corresponds to the study plan. Teachers choose the topic of an OL class by themselves, taking into account the areas in which he/she can demonstrate improvements, test new techniques and methods, and carefully examine the progress of students' learning at

different stages of the lesson. When preparing for an OL class, the teacher takes care to use the latest information, select materials from pedagogical, scientific, technical, and methodological literature, and apply the results of feedback from visitors to form their future teaching.

The OL class starts in the following way: The invited staff enters the class before the bell rings, takes pre-prepared seats, completes special forms, and obtains the lesson plan prepared by the teacher who is teaching the lesson. All of the invited staff should observe the pedagogical approaches aligned with the OL goals and make notes about the lesson without interfering with what goes on in the class and without expressing their observations or interpretations to both the teacher and the students in the class. The following questions are often used to guide the observation: Is the teacher achieving his/her goal? What are the effective pedagogical strategies used to meet the OL goal(s)? What are the results of his/her teaching strategies on students' learning? The results of the observations are reflected in the OL observation form [20]. The structural representation of OL is shown in Figure 1.

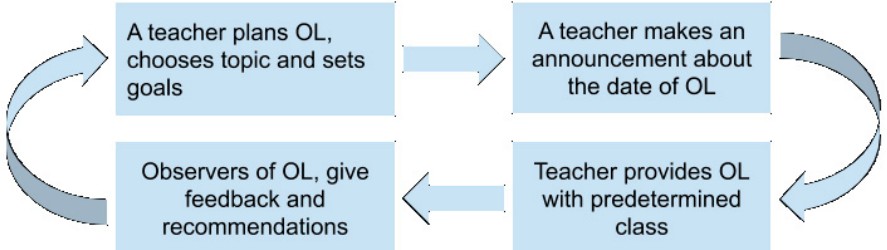

**Figure 1.** Structural diagram of OL.

*2.3. Evaluation of Open Lesson*

After finishing a lesson, an analysis of the lesson is conducted with the observers (e.g., teachers and administrators). First, the instructor of the lesson speaks. He/She recalls his/her OL goals and provides a reflection of his/her teaching approaches, focusing on what they did and did not find success in. The following question guides this debriefing process: Are the goals of the OL achieved and what are some explanations for the successes and challenges? Visitors then have the opportunity to provide feedback on the OL, such as providing a thorough analysis of the merits and weaknesses of the lesson, an assessment of the activities for supporting students' learning, and the degree to which the goals were achieved. During the discussion, it is possible to note the shortcomings and mistakes made in the organization and content of the OL, and provide recommendations on improving this process. The main criterion for assessing the effectiveness of an OL should be the quality of knowledge presented, teaching skills, and student understanding.

In the end, the head of the methodological department and the representative of the administration provide concluding comments. They summarize the comments and note if any compulsory parts of the OL were missed, give an assessment of the used methods, note the depth of disclosure of the methodological goal, and make a conclusion about the expediency of using the presented experience. After this speech, the teacher who conducted the OL class gives his/her own response to these comments. He/She notes which comments he/she accepts, and those with which he/she does not agree and why [21]. The decision of the pedagogical council can then include specific proposals and recommendations to continue making improvements to teaching.

*2.4. Research on Open Lesson*

Existing studies focus primarily on the implementation of the pedagogical approaches in the study processes. For example, studies [8,9] examined the results of OL focused on new technology use for supporting students' grammar skills in high school English classrooms. Another study using OL was conducted on the effectiveness of using elements of critical thinking in group work [10]. Besides CISC, related research in OL is provided

in some other countries; the authors of [3,4] described the stages and procedures of OL in China. Another application of OL for pre-service teachers was implemented in Macao [12]. In a study conducted in Japan by the authors of [2], a case of OL was presented and examined. Similarly, the authors of [11] systematically presented the practice of OL and argued its consequences for American teaching.

### 2.5. Implementation of Open Lesson in Other Countries

The practice known as open class is also popular in China [4]. Open class is a regular and expected part of daily school life for Chinese teachers. The purpose of this form of open class is to provide teachers with the opportunity to observe and discuss subject content and pedagogical practices within the immediate context of their own school or classroom; in other words, teachers learning from each other.

Japanese open lesson is also very similar to open lesson in Kazakhstan. The features of Japanese open lesson are: a large number of teachers observe a lesson taught by a colleague and discuss various aspects of it right after the lesson. Besides teachers, there may also be other guests, such as university researchers, who participate in the observation of the lesson and in the discussion afterwards.

The European teacher training development activity is called demonstration classroom. The purpose of demonstration classroom is to support Standards for Professional Learning. In demonstration classroom, teachers observe a colleague's lessons and, afterwards, actively participate in a collaborative debriefing session with an instructional guide and the demonstration classroom teacher to ask specific questions, share ideas, and plan for the implementation of the observed effective teaching strategies with the support of the guide (coach or instructional leader) in their own classrooms.

### 2.6. A Comparison of Open Lesson and Lesson Study

In some aspects, OL is similar to other kinds of teacher professional development activities used in other countries. One of the types of such professional development activities is Japanese lesson study (LS) [22,23]. Japanese LS is an approach to professional learning characterized by research in action through observations of common lessons [24]. As mentioned by the authors of [2], for Japanese teachers, there are no significant differences between OL and LS. CISC OL and Japanese LS provide a similar format. The basic idea of these two activities is that visiting teachers from other schools observe a class that is taught by a teacher of the school who, after the class, participates in a discussion with the visiting teachers and sometimes other invited experts (university teachers, school administration members, etc.) on the details of the lesson.

In contrast, the clear distinction between OL and LS can be found when we compare Kazakh OL or CISC OL with American LS. American LS involves a cyclic practice of professional development in which a group of teachers jointly implement, plan, teach, monitor, and analyze teaching practices and students' learning, and document their findings. During American LS, both experienced and novice teachers can innovate or improve their pedagogical approaches [25]. In this paper, we mainly compared CISC OL in general [18] with American LS [1,26] to show the differences and similarities of two types of lesson activities, which is shown in Table 1.

**Table 1.** A comparison of OL and American LS.

| Characteristics of Activity | Similarities | Differences |
|---|---|---|
| Defining the problem | In OL and LS, problems are defined by overall knowledge indicators (marks, attitudes, etc.) of students in certain classes [25]. | LS is a students' problem-solving process. Teachers shape and focus the problem until it can be addressed by the specific classroom lessons with particular teaching methods [1]. In contrast, the OL problem is defined according to the teacher's teaching methodology, knowledge, and attitude, which can be reflected in the students' performance at school in the future [18]. |
| Planning the Lesson | Both methods aim to set a goal, investigate the problem, define standards of activity, and define the date(s), and planning can be discussed with other experts [22]. | LS is planned by a group of teachers. After defining problems, observing teachers define "focusing groups." Focusing groups are groups of students who have certain problems with study [26]. OL is planned by a teacher, who prepares all materials for the lesson [10]. |
| Teaching the Lesson | A particular teacher guides the lesson, and other experts observe it. During the lesson, various types of teaching techniques, evaluation methods, etc., can be used according to the pre-determined lesson plan. | In LS, during the lessons, observers mainly focus on students' actions. They observe the efforts of students in "focus groups" for further discussion and analyze the problems, which were noticed during the lesson. By this, they define why students have such kind of problems and try to create a new teaching technique to eliminate these problems in the second cycle of LS [25]. During OL, invited guests mainly focus on the teaching methodology of a teacher, and they observe the availability of given material to students and pay attention to students' actions and answers during the lesson. |
| Evaluating the Lesson and Reflecting on its Effects | There are group meetings soon after the lesson. Teacher and observers discuss the lesson's procedure and share an opinion about the lesson's progress. Observers suggest a new idea for further development of the lesson [3]. | Teachers discuss the lesson's progress and talk about focus groups' students just after LS. They identify the main problems that "focus groups" were encountered during the lesson and try to come up with the solution to the problem. The second cycle of lesson study is provided by taking into account these suggestions. Observers give valuable comments to the teacher to improve the method of teaching OL. |
| Revising the Lesson | According to the gained data, reflections, and advice of observers, an instructor provides a new cycle of the lesson. Revised lessons are taught taking into account previous recommendations. An instructor can change the materials and activities in the second cycle of the lessons [4]. | The second cycle of LS can be provided with other classes to make a comparison of teaching methods with different classes or, more often, another member of the group teaches the revised LS. They can try the same teaching method in various subjects and observe the progress of "focus groups." Revised OL is taught by the same teacher taking into account previous recommendations. |
| Sharing the Results | Teachers announce the results of the lesson from OL and LS in their school portfolio, journals, magazines, etc. | Sharing the results could be completed in any form in LS and OL. There is no difference in the results sharing of OL and LS. |

*2.7. Importance of Individuals' Attitudes and Beliefs*

For the successful implementation and long-term sustainment of professional development practices, such as OL, both the instructors and students must believe these

pedagogical practices have meaningful and positive impacts on learning [3]. In this study, we define attitudes as a predisposition or mental state of readiness towards a particular set of practices that consequently influences the subsequent interpretation of information and behaviors [27]. We also examine beliefs, a related but distinct construct defined as psychologically held understandings about particular ideas and outcomes of behaviors [28].

Studies have shown that college professors' attitudes and beliefs about teaching techniques strongly predict their instructional choices, such as the degree to which they take up and implement active learning strategies [29]. Similarly, students' attitudes towards their college instructors' instruction also play an important role in how instructors teach, as well as how students engage in the learning process. For example, studies have shown that students who have positive attitudes towards their learning experiences, coupled with beliefs in their ability to do well in their courses, support persistence during difficult tasks and achievement [28,30]. Notably, models of professional learning underscore the dynamic nature of individuals' attitudes and beliefs, as educators and learners participate in new teaching and learning practices, reflect on the effectiveness of these approaches, and use the evidence from the classroom to inform future approaches [6]. OL provides a unique context to examine both college instructors' and students' attitudes and beliefs as they relate to new teaching strategies.

## 3. Materials and Methods

### 3.1. Research Design

Both quantitative and qualitative approaches were utilized to investigate the effects of gender, discipline, nationality, and teaching experience on teachers' and students' attitudes and beliefs about OL. Mixed methods research is known as the type of research where the quantitative and qualitative research procedures, methods, approaches, concepts, or language are combined into a sole study [31]. Data were collected from teachers and students in the first semester of the 2017–2018 academic year using two different surveys. The quantitative data collected from the surveys were analyzed using SPSS (SPSS 21.0 for Windows, SPSS Inc., Chicago, IL, USA) software, and participants' written responses were analyzed qualitatively [32] to gauge participants' further attitudes and beliefs about OL.

### 3.2. Participants

Seventy-five teachers from five different cities in Kazakhstan who taught at the university level participated in this study. Participants were contacted via email and phone. The aim of the study was explained and the survey was sent to those who agreed to participate in our study. Some of the involved teachers also agreed to administer the survey to their students.

For the teachers: First, the mean years of teaching experience for the teacher participants was 18.29. Second, 64% of teachers were women and 36% were men. Third, teachers taught in 21 different disciplines (59% social sciences and 41% natural sciences). Fourth, 64% of the teachers were Kazakh, 20% were Russian, 7% were Korean, and 9% were others (Azerbaijani, Tatar, Uzbek, and Kyrgyz). Fifth, the mean number of OL conducted by the teachers was 9.35. Sixth, the mean number of OL the teachers participated in was 11.77 across their whole career.

Two hundred and fifty university students in various grades from three different cities in Kazakhstan also participated in this study. Students at various socioeconomic levels were from both state and private universities. The demographic data for students showed: First, the mean age of the students was 18.06. Second, the mean GPA of the students was 4.33 (about half of the students did not state their GPA, so it is possible that low GPA students did not express theirs). Third, 56% of students were female and 44% were male. Fourth, students were studying 25 different disciplines (66% social sciences and 36% natural sciences). Fifth, 66% of the students were Kazakh, 11% were Russian, 7% were Korean, 5% Uyghur, 3% Tatar, and 8% were others (Azerbaijani, Chechen, Dungan, Italian, Kurd, Kyrgyz, Tatar, Tajik, Turk, and Uzbek). Sixth, the mean number of OL that the students participated in was 3.74 across their whole educational life.

### 3.3. Instruments

Separate data collection instruments were developed by the researchers for teachers and students to gather data to supply responses to the research questions. In selecting the most appropriate survey items, tools investigating the teachers' and students' perceptions, attitudes, and beliefs about Japanese lesson study were examined [13–15].

The appropriate primary validity evidence for the developed instruments was content validity. To improve the content validity of the tools, the four authors carefully reviewed all items of the instruments. The tools were emailed between the authors several times to avoid unclear directions, vague items and words, and unnecessary items. The final version was slightly amended by two experts from educational sciences who had experience with survey research. There were 25 items on the teachers' survey and 15 items on the students' survey along with some additional demographic items. All Likert-type items can be seen in the results section, where their descriptive statistics are presented. The items were rated on a five-point Likert-type scale used in this study, where 5 corresponds to strongly agree, 4 to agree, 3 to neutral, 2 to disagree, and 1 to strongly disagree. Hence, a score below 3 on this scale indicates a negative, a score of 3 indicates a neutral, and a score above 3 indicates a positive attitude and belief.

The dimensions for the teacher survey were: attitudes and beliefs about OL and teaching (14 items), attitudes and beliefs about OL and students' learning (6 items), and attitudes and beliefs about OL and professional collaboration with colleagues (5 items). The dimensions for the student survey were: attitudes and beliefs about OL and teaching (7 items) and attitudes and beliefs about OL and learning (8 items).

Reliability of the instruments was established using the Cronbach's alpha internal consistency reliability estimates. The overall reliability estimated for the teacher instrument was 0.85 while the overall reliability estimated for the student instrument was 0.82. One item from the teacher survey had a negative item–total correlation and we removed this question. After this item was removed, the reliability coefficient for the teacher survey raised to 0.86.

### 3.4. Data Collection and Data Analysis

Some of the teacher data were collected via email, while most of the data were collected by the researchers through paper-and-pencil survey sheets administered to each participant. Teachers either responded to the surveys before or after a meeting or during their spare time. The survey was administered to students by their course teachers toward the end of their courses. The surveys were anonymous. All participation in the surveys was voluntary and no incentives were offered for student participants. A total of 75 teachers and 250 students completed the surveys. After the data cleaning process, the results of 72 teachers and 239 students were reported.

The teacher data and student data were analyzed separately. Initially, the descriptive statistics were conducted for both sets of data, followed by the inferential statistics; finally, some qualitative results were reported. First, histograms, P-P plots, and scatterplots were examined to ensure that the assumptions of parametric tests were met. Mean score statistics for gender, discipline, nationality, and teaching experience across the dimensions of the teacher and student surveys were calculated. Moreover, mean score statistics and percentages of agreement (1 and 2), neutrality (3), and disagreement (4 and 5) for each item were also calculated. Using the SPSS program, a two-way multivariate analysis of variance (MANOVA) was conducted for both teacher and student responses to determine the effects of the independent variables on the dependent variables. The qualitative data provided by both teachers and students (for the purpose of answering the third research question) were sorted simply: firstly categorized on the advantages and disadvantages of OL and secondly categorized according to common responses.

## 4. Results

### 4.1. Analysis of Teacher Data

Descriptive statistics: Teachers' averages on the three levels of the survey are indicated in Table 2. The mean overall scores were 3.33 on attitudes and beliefs about OL and teaching, 3.48 on attitudes and beliefs about OL and students' learning, and 3.80 on attitudes and beliefs about OL and professional collaboration with colleagues. These values indicate that teachers with different genders, disciplines, nationalities, and teaching experience reported weak positive attitudes and beliefs with respect to three dimensions of the teacher survey.

**Table 2.** Mean score statistics for teachers.

|  |  | Teaching | Learning | Collaboration |
|---|---|---|---|---|
|  | All | 3.33 (0.41) | 3.48 (0.47) | 3.80 (0.48) |
| Gender | Female | 3.39 (0.39) | 3.55 (0.36) | 3.90 (0.34) |
|  | Male | 3.21 (0.42) | 3.36 (0.60) | 3.61 (0.63) |
| Discipline | Natural Science | 3.38 (0.45) | 3.46 (0.56) | 3.79 (0.41) |
|  | Social Science | 3.29 (0.37) | 3.49 (0.40) | 3.80 (0.53) |
| Nationality | Kazakh | 3.33 (0.42) | 3.45 (0.50) | 3.78 (0.56) |
|  | Russian | 3.41 (0.36) | 3.63 (0.33) | 3.80 (0.34) |
|  | Other | 3.20 (0.42) | 3.39 (0.43) | 3.85 (0.30) |
| Teaching experience | 1–10 | 3.33 (0.58) | 3.35 (0.64) | 3.83 (0.48) |
|  | 11–20 | 3.21 (0.36) | 3.45 (0.43) | 3.70 (0.68) |
|  | Over 20 | 3.41 (0.24) | 3.60 (0.29) | 3.84 (0.28) |

Several notable statistics are presented in Table 2. First, for all independent variables, teachers' survey scores increase from teaching to collaboration. In other words, when compared, teachers' total score for the effect of OL on teaching is the lowest, and the effect on collaboration is the highest. Second, female teachers have more scores on all levels of the survey when compared to male teachers. Third, teachers in natural and social sciences have very similar scores on the three dimensions of the survey. Fourth, Russian teachers have more scores on teaching and learning, while teachers other than Kazakh and Russian have the lowest score on teaching and the highest score on collaboration. Fifth, teachers with experience over 20 years have more scores on all levels. However, these statistics are limited in providing information about general trends in the data.

In scoring the teachers' attitudes and beliefs about OL, responses for 1 and 2 were grouped together because they both represent disagreement. Responses for 4 and 5 were also grouped together because they both represent agreement. We can categorize the level of attitudes and beliefs as positive—70% and over, weak positive—40–70%, and negative—below 40% [33] (Balta, Yerdelen-Damar, and Carberry, 2017). Four of the items (6, 8, 9, and 14) revealed students to have negative attitudes and beliefs, but twelve of the items (3, 4, 5, 10, 11, 13, 15, 17, 18, 21, 23, and 24) displayed weak positive attitudes and beliefs compared to eight of the items (1, 2, 7, 12, 16, 20, 22, and 25) displaying positive attitudes and beliefs. On the item basis, Item 25, referring to 'I implement the best practices I see in OL', was rated highest (M = 4.15), and Item 6, referring to 'OL is too time-consuming', was scored the lowest (M = 1.90).

Inferential statistics: Using the SPSS Program, we conducted the two-way MANOVA to assess the effect of gender and discipline on three levels of teachers' attitudes and beliefs about OL. Assumptions of MANOVA—normality, independence of observations, homogeneity of covariance matrices of each group, and the random and independent sampling from the population—were tested. The Kolmogorov–Smirnov test result was significant in all dimensions of teacher the survey, indicating non-normal distributions. Box's test of equivalence of covariance matrices showed that the observed covariance matrices of the dependent variables are equal across groups ($p = 0.346 > 0.01$). Pillai's trace

was utilized for the analysis of MANOVA because the normality assumption was not met and it is more robust to violations [34].

The null hypothesis for this MANOVA is that the four independent variables (gender, discipline, nationality, and teaching experience) are equal to the three dependent variables (teaching, learning, and collaboration) for teachers' scores. The results for the two-way MANOVA indicated a non-significant main effect for gender (Pillai's trace = 0.35, $F_{(3, 66)}$ = 0.79, $p > 0.05$), a non-significant main effect for teaching experience (Pillai's trace = 0.90, $F_{(6, 90)}$ = 0.71, $p > 0.05$), a non-significant main effect for nationality (Pillai's trace = 0.22, $F_{(6, 90)}$ = 1.92, $p > 0.05$), and a non-significant main effect for discipline (Pillai's trace = 0.16, $F_{(3, 44)}$ = 2.72, $p > 0.05$). These results suggest that we cannot reject the null hypothesis and we accept that there is no difference between gender, nationality, and discipline as well as between teaching experience in terms of their attitudes and beliefs about OL.

Analysis of qualitative data: Along with Likert-type items, there were several open-ended items in the survey to further reveal teachers' thoughts about OL. First, teachers declared that it takes them 1–2 days to prepare for an OL class. Second, they stated that approximately 3–4 teachers participate in their OL implementations. Third, teachers responded that before an OL class, they generally review the topic that they are going to teach during the OL class, and prepare handouts for the OL class. Fourth, teachers specified several advantages of OL for their teaching and for their students' learning, such as the OL increases the experience of teaching, helps to increase knowledge in the field of pedagogical content, and improves the quality of teaching. Teachers also noted that OL allows the teacher to mobilize his/her best skills and get an assessment of his/her pedagogical skills from colleagues. They also stated that an OL unites the teaching staff in a professional way; they allow for a better understanding of each other. Some examples of the teachers' responses are: "An unconventional method of communicating with the audience", "Increasing teacher responsibility, expanding teaching methods", "Using interactive teaching methods", "Students are motivated during an open lesson", "Helps to improve the quality of teaching", "Looking from the side and valuable advice from colleagues", and "Opportunity to reveal yourself".

Fifth, teachers also identified several disadvantages of OL, such as preparation for the OL class takes too much time. The attitude towards holding an OL class for teachers is often formal when conducting an OL class and the emphasis is usually placed on the external attributes of the quality supply of the pedagogical material, while the quality of the course should depend on the internal dynamics. The evaluation of colleagues is not always objective (one can unreasonably give a negative assessment, especially when an open lesson class is attended by incompetent teachers). Some answers from teachers are: "The quality of the lesson is higher than the usual level of formal assessment by teachers", "The determining factor is the relationship with colleagues, and sometimes, they are formal in nature", and "Amount of time".

### 4.2. Analysis of Teacher Data

Descriptive statistics: The total mean score determined from the student survey was 3.41 points (SD = 0.41), on a scale from 1 to 5, with 1 being the most negative attitude and belief toward OL and 5 the most positive.

Table 3 shows that: (a) All independent variables of students' attitudes and beliefs about OL increase from teaching to learning. In other words, when one is compared to the students' total scores on the effect of OL on teaching, one is less than its effect on learning. (b) Gender differences on both levels of the survey are nearly zero. (c) Students learning social sciences have stronger positive attitudes and beliefs when one is compared to students learning natural sciences. (d) Kazakh and non-Kazakh students have similar attitudes and beliefs about the effect of OL on teaching, while Kazakh students have stronger positive attitudes and beliefs on the effect of OL on learning.

**Table 3.** Mean score statistics for students.

|  |  | Teaching | Learning |
|---|---|---|---|
|  | All | 3.30 (0.43) | 3.52 (0.57) |
| Gender | Female | 3.33 (0.41) | 3.52 (0.62) |
|  | Male | 3.26 (0.45) | 3.52 (0.51) |
| Discipline | Natural Sciences | 3.23 (0.45) | 3.36 (0.53) |
|  | Social Sciences | 3.33 (0.42) | 3.61 (0.57) |
| Nationality | Kazakh | 3.29 (0.41) | 3.56 (0.56) |
|  | Other | 3.30 (0.46) | 3.42 (0.59) |

Students' attitudes and beliefs about OL indicate that none of the items revealed students to have positive attitudes and beliefs but twelve of the items (1, 2, 3, 4, 5, 9, 10, 11, 12, 13, 14, and 15) displayed weak positive attitudes and beliefs compared to three of the items (6, 7, and 8) displaying negative attitudes and beliefs. On the item basis, of the 15 items that comprise the instrument, the item with the highest mean is 'OL is boring'; however, since the score was calculated after reversing, it turns out to be 'OL is not boring' (M = 3.72; SD = 1.20). The item with the lowest mean is 'OL is too time-consuming' (M = 2.76; SD = 1.05).

Inferential statistics: The null hypothesis for this MANOVA is that the three independent variables (gender, discipline, and nationality) are equal to two dependent variables (teaching, learning) for students' scores. The results for the two-way MANOVA indicated a non-significant main effect for gender (Pillai's trace = 0.018, $F_{(2, 222)} = 2.02$, $p > 0.05$), a non-significant main effect for nationality (Pillai's trace = 0.004, $F_{(2, 222)} = 0.48$, $p > 0.05$), and a significant main effect for discipline (Pillai's trace = 0.052, $F_{(2, 222)} = 6.10$, $p < 0.05$).

Because a significant main effect was observed for discipline, we examined the differences among disciplines for two subscales of the survey. ANOVA tests between subjects were employed to test the effects of discipline on learning and teaching dimensions of attitudes and beliefs of students. The $\alpha$ level was adjusted to prevent committing a Type I error using Bonferroni's correction in which the level is divided by the number of dependent variables [35]. The adjusted alpha value for ANOVA analysis was reduced to $\alpha$ reduced = 0.025 because there were two dependent variables. No significant effect of discipline on the teaching dimension was revealed for the adjusted level, while there was a significant effect of discipline on the learning dimension ($F_{(1, 2.40)} = 11.79$, $p < 0.025$). This shows that, compared to students learning natural sciences (M = 3.02, SD = 0.48), students learning social sciences (M = 3.27, SD = 0.44) have significantly stronger positive attitudes and beliefs about the effect of OL on learning.

Analysis of qualitative data: Along with Likert-type items, there were several open-ended items in the survey to further reveal students' thoughts about OL. First, they stated that during OLs, teachers used different methods to attract students' interest. Teachers explained everything more clearly, explained the topics in more detail, prepared for the lesson better, made the lessons more interesting, conducted the student-centered lessons, approached the OL with the great enthusiasm, and used more technology, while students' participation in lessons increased. Second, students expressed several advantages of OL, such as teachers carefully prepared for OL, courses were informative and planned, teachers used new methods that were very interesting, and students were more involved during OL. Some of the students' answers were: "Students become more interested in this subject, and people from outside can tell how best to conduct classes", "The audience is more disciplined and engaged in work", "Everybody comes ready for classes", "Thanks to an OL, students can demonstrate their ability to speak, listen attentively, learn to work in groups, students prepare themselves more diligently for a lesson if it is open", "All students work", "Repetition and consolidation of the topics covered", "Detailed Understanding", "Teamwork", "More discussion", and "Opportunity to prove oneself".

Third, students expressed several disadvantages of OL, such as that it is not always possible to express their opinion and there is a fear of public speaking and of criticism from others, as well as feelings of excitement and tension. Some students' responses: "Stress, agitation, psychological crush", "Fear of speaking out", "The disadvantage is the time limit when the teacher does not fully reveal", "Many have a fear of the audience and the fear of the scene, because of this the person does not work at 100%", "There are no shortcomings", "Rigid requirements for evaluation", "Cannot give time to each student", and "Resource intensity, it takes a lot of time, takes time, time".

## 5. Discussion

Regarding the first research question, we found slightly positive attitudes and beliefs about OL from both teachers and students. The weak positive attitudes of teachers can be attributed to the fact that OL is mandatory for teachers. However, students' weak positive attitudes need to be examined in future studies. OL conducted by inexperienced teachers and the frequent implementation of OL may be two possible reasons.

For the second research question, we conducted factorial MANOVA to analyze the effect of gender, discipline, nationality, and teaching experience on teachers' attitudes and beliefs, which yielded no group differences. One possible reason for this finding is that OL is extensively implemented and has a long tradition in Kazakhstan as well as nearby countries. Students are subjected to OL applications starting in primary school and teachers start facing OL activities during undergraduate education.

Similarly, we conducted factorial MANOVA to analyze the effect of gender, discipline, and nationality on students' attitudes and beliefs, which yielded only a significant difference in discipline. Students learning social sciences demonstrated stronger positive attitudes and beliefs about the effect of OL on learning when compared to students learning natural sciences. To clarify this result, further research is needed because no difference was observed for the teachers in social and natural sciences while there is a significant result for students. That is, in future research, OL implementations in social sciences classes and natural sciences classes should be independently observed and results should be reported. However, we shared our results with several teachers from both social and natural sciences, and they explained this difference as follows:

Students and teachers in OL, in social sciences, feel free to interpret the context of the given topic; they can add their own opinion to describe art, stories, etc. In social sciences, it is more likely that a diverse range of opinions and interpretations are generally encouraged and valued. In natural sciences, the situation is more likely to be stressful during the OL. A pupil should show their ability to solve problems and the teacher should find some new way to explain natural phenomena. This needs more time and the power differentials between the instructor and the pupils are more salient.

Regarding the third research question, open-ended items in the survey were used to further reveal participants' thoughts about OL. Teachers see OL as a means to their professional development while students see OL as a means of quality education. On the other hand, teachers found that preparation for OL was challenging and some students criticized the environment of OL as not being natural because of the presence of other teachers in the class.

The positive effect of OL on observing teachers was both stated by our participants and by Chinese participants from Liang's [3] study. Many teachers specified that they learn a lot when observing other teachers conducting OL and that participating in OL assisted them in improving their own teaching. Similarly, teachers from both studies stated that OL broadens their visions of effective classroom teaching. The effect of participation in open lesson gave similar results in ours and the study by Sun et al. [12]. Namely, OL enables teachers to understand students' learning problems, observe real classroom situations, set clearer teaching goals, and expand didactic reasoning through comparison and judgment.

Even though there is no exterior motivation for the practice of OL, both Kazakh and Japanese teachers [2] find this practice valuable and professionally satisfying.

## 6. Conclusions and Implications

We are expecting that this study on OL will lead more researchers to focus on this topic. The current study was conducted in Kazakhstan; more studies are needed to see if it is similar in other CISCs. Moreover, the study has been restricted and applied only for undergraduate students and additional research is needed to see if findings can be generalized for primary and high schools. We hope that this research provides a foundation for other studies and pursuits into the open lesson practice in Kazakhstan.

Perhaps, the key implication of the results from the present study is that researchers and educators need to consider adaptations to the OL implementations to increase teachers' and students' weak positive attitudes and beliefs to higher levels. Students learning natural sciences demonstrated less than positive attitudes and beliefs about the effect of OL on learning when compared to social science students. This means that natural science teachers should participate in OL lessons conducted by social science teachers to see their in-class implementations.

We suggest that educational experts compare OL to Japanese LS and restructure the OL. We also think that this model of PD is worth being considered and attempted in many countries' in-service training for teachers. Furthermore, teachers' and students' responses to open-ended questions revealed that participating teachers benefit more when compared to the teachers conducting OL. This implies that novice teachers should be encouraged or necessitated to participate in OL presented by experienced teachers. In other words, conducting OL is mandatory for teachers; participation should also be mandatory for novice teachers.

**Author Contributions:** Data curation, N.B.; methodology, N.J. and N.B.; writing—original draft preparation, M.A. and N.B.; writing—review and editing, C.L.B.; supervision, N.B.; project administration, A.K. All authors have read and agreed to the published version of the manuscript.

**Funding:** This research received no external funding.

**Institutional Review Board Statement:** The study was conducted in accordance with the Declaration of Helsinki, and approved by the Institutional Review Board of Suleyman Demirel University (#58, 18.08.2022).

**Informed Consent Statement:** Informed consent was obtained from all subjects involved in the study.

**Data Availability Statement:** Data will be provided upon request. Any reader can request the data.

**Conflicts of Interest:** The authors declare no conflict of interest.

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
