# Peer review of "Open Lesson as a Means of Teachers’ Learning"

_education, doi:10.3390/educsci12100692_

Round 1

Reviewer 1 Report

Dear authors, thanks for your contribution. The novelty aspect dealt with in the context of open lessons sounds compelling. I am wondering how open an open lesson can be, given certain societal or political circumstances in the specific sampling context? In other words, how do you consider possibly existing biases or phenomena such as social expectancy into your rationale? Would you think that it might have an impact on beliefs and attitudes of teachers and students towards OL? Please further explain conditions and drivers contributing to the teaching-learning system.

Author Response

Dear reviewers

Thank you for your recommendation to improve our work. We did following changes of the manuscript and highlighted them the text with yellow color.

Reviewer’s comment:

Please revise the title so that it would suit the theme of our special issue.
We herby suggest the following title: Open Lesson as a means of teachers'
learning

Authors response:

We have changed the title of the article to Open Lesson as a means of teachers'
learning

Reviewer’s comment:

Please discuss OL as a vast international practice and not as a
local one.

Authors response:

We have added one chapter to the manuscript about Implementation of Open lessons in other countries.

Reviewer’s comment:

We think you should omit the students' perspective (Unless you
can prove its value to the development of teachers).

Authors response:

The views of students on the implementation of the OL are needed for policymakers. We want to keep students' perspectives in our research.

Reviewer’s comment:

Please add the aims and questions of the research after the Literature Review.

Authors response:

The aims and research questions are written and highlighted in the text.

Reviewer’s comment:

Please add the ethics you applied.

Authors response:

The ethics is added

Reviewer 2 Report

Comments for Education Sciences 1883304

I have some concerns about the professional writing style and am not sure if it fits this journal.  Should “we” be used?  Should abbreviations be at the beginning of sentences?  There are also many English language grammar errors, most common is subject-verb plurality agreement.

I have read the opening paragraphs and do not understand what Open Lesson is.  The literature review leads me to think it possibly a peer review of teaching process.

Were the people surveyed making or watching Open Lessons?

Data looks useful but I can’t tell what OL is so I just can’t appreciate the full manuscript.  A rework of introduction and literature review would be very helpful.

Author Response

Dear reviewers

Thank you for your recommendation to improve our work. We did following changes of the manuscript and highlighted them in the text with yellow color.

Reviewer’s comment:

Please revise the title so that it would suit the theme of our special issue.
We herby suggest the following title: Open Lesson as a means of teachers'
learning

Authors response:

We have changed the title of the article to Open Lesson as a means of teachers'
learning

Reviewer’s comment:

Please discuss OL as a vast international practice and not as a
local one.

Authors response:

We have added one chapter to the manuscript about Implementation of Open lessons in other countries.

Reviewer’s comment:

We think you should omit the students' perspective (Unless you
can prove its value to the development of teachers).

Authors response:

The views of students on the implementation of the OL are needed for policymakers. We want to keep students' perspectives in our research.

Reviewer’s comment:

Please add the aims and questions of the research after the Literature Review.

Authors response:

The aims and research questions are written and highlighted in the text.

Reviewer’s comment:

Please add the ethics you applied.

Authors response:

The ethics is added

Reviewer 3 Report

Please, find attached the document with comments and suggestions.

Author Response

(The authors gave the same response as above.)

Round 2

Reviewer 2 Report

The revision is much clearer in describing the practice and value of open lesson.